# Human Trabecular Meshwork (HTM) Cells Treated with TGF-β2 or Dexamethasone Respond to Compression Stress in Different Manners

**DOI:** 10.3390/biomedicines10061338

**Published:** 2022-06-06

**Authors:** Megumi Watanabe, Tatsuya Sato, Yuri Tsugeno, Araya Umetsu, Soma Suzuki, Masato Furuhashi, Yosuke Ida, Fumihito Hikage, Hiroshi Ohguro

**Affiliations:** 1Departments of Ophthalmology, School of Medicine, Sapporo Medical University, Sapporo 060-8556, Japan; watanabe@sapmed.ac.jp (M.W.); yuri.tsugeno@gmail.com (Y.T.); araya.umetsu@sapmed.ac.jp (A.U.); ophthalsoma@sapmed.ac.jp (S.S.); funky.sonic@gmail.com (Y.I.); fuhika@gmail.com (F.H.); 2Departments of Cardiovascular, Renal and Metabolic Medicine, Sapporo Medical University, Sapporo 060-8556, Japan; satatsu.bear@gmail.com (T.S.); furuhasi@sapmed.ac.jp (M.F.); 3Departments of Cellular Physiology and Signal Transduction, Sapporo Medical University, Sapporo 060-8556, Japan

**Keywords:** 3D spheroid culture, human trabecular meshwork (HTM), dexamethasone, TGF-β2, compression stresses

## Abstract

To characterize our recently established in vitro glaucomatous human trabecular meshwork (HTM) models using dexamethasone (DEX)- or TGF-β2-treated HTM cells, (1) two-dimensional (2D) cultured HTM cells were characterized by means of the real-time cellular metabolism analysis using a Seahorse analyzer, and (2) the effects of mechanical compression stresses toward the three-dimensional (3D) HTM spheroids were evaluated by analyzing the gene expression of several ECM proteins, inflammatory cytokines, and ER stress-related factors of those 3D HTM spheroid models. The results indicated that (1) the real-time cellular metabolism analysis indicated that TGF-β2 significantly induced an energy shift from mitochondrial oxidative phosphorylation (OXPHOS) into glycolysis, and DEX induced similar but lesser effects. In contrast, ROCK2 inhibition by KD025 caused a substantial reverse energy shift from glycolysis into OXPHOS. (2) Upon direct compression stresses toward the untreated control 3D HTM spheroids, a bimodal fluctuation of the mRNA expressions of ECM proteins was observed for 60 min, that is, initial significant upregulation (0–10 min) and subsequent downregulation (10–30 min) followed by another upregulation (30–60 min); those of inflammatory cytokines and ER stress-related factors were also bimodally changed. However, such compression stresses for 30 min toward TGF-β2- or DEX-treated 3D HTM spheroids induced downregulation of most of those of inflammatory cytokines and ER stress-related factors in addition to upregulation of COL1 and downregulation of FN. The findings presented herein indicate that (1) OXPHOS of the HTM cells was decreased or increased by TGF-β2 or DEX stimulation or ROCK2 inhibition, and (2) mechanical compression stresses toward 3D HTM spheroids may replicate acute, subacute, and chronic HTM models affected by elevated intraocular pressures.

## 1. Introduction

Based upon the evidence-based medicine related to the glaucomatous optic neuropathy (GON), hypotensive therapies to suitably maintain the intraocular pressure (IOP) by either an antiglaucoma medication, laser treatment, or surgery have been established so far [1,2,3,4]. As for the homeostasis for the maintenance of the IOP levels, precisely regulated balance between production and drainage through the trabecular meshwork (TM)/Schlemm’s canal route and the uveoscleral route [5] of the aqueous humor (AH) is extremely important. In terms of the responsible etiology inducing the elevation in the IOP, the increase in the AH outflow resistance caused by the large excess of deposit of extracellular matrix (ECM) proteins on the TM appears to be a key mechanism for both primary open-angle glaucoma (POAG) as well as steroid-induced glaucoma (SG) [6]. In fact, several in vivo and in vitro studies have demonstrated that excess deposition of ECM proteins could be induced by transforming growth factor β2 (TGF-β2) as well as glucocorticoids, resulting in IOP elevation [7,8,9,10].

For the study purpose to elucidate POAG and SG pathogenesis in detail, several in vitro cell cultures using human TM (HTM) have recently been utilized in conjunction with TGF-β2 and dexamethasone (DEX) to establish suitable models replicating glaucomatous TM properties [8,9,10,11]. However, since most of these studies using conventional 2D cell cultures were quite different, with the multiple sheets-structured HTM [12], we developed a more relevant in vitro model using a unique 3D cell culture method [13,14,15] to overcome such a disadvantage. In fact, our prepared 3D HTM spheroids became significantly and differently smaller and stiffer in response to TGF-β2 or dexamethasone (DEX) stimulation [16,17]. Furthermore, these glaucomatous 3D HTM models were substantially and beneficially affected by the presence of antiglaucoma drugs including Rho-associated coiled-coil-containing protein kinase (ROCK) inhibitors and others [16,18,19]. Therefore, based upon these collective findings, we rationally suggested that our established 3D HTM spheroids treated by TGF-β2 or DEX may become a physiologically relevant in vitro model replicating a POAG or SG TM, respectively. However, they were still insufficient in terms of not only biological properties of both 3D HTM spheroid models themselves, but also, as an additional important factor, of the effects of elevated mechanical stress to mimic IOP elevation. 

In the current study, to study these unidentified issues, TGF-β2- or DEX-treated 3D HTM models (as above) were further analyzed using a real-time cellular metabolism analyzer, and effects of the mechanical compression stresses toward these models were also investigated.

## 2. Materials and Methods

All experimental procedures using human tissue/cells were conducted as follows in compliance with the tenets of the Declaration of Helsinki and approved by the internal review board of Sapporo Medical University. 

### 2.1. 2D and 3D Cultures of HTM Cells

2D and 3D spheroid cultures of HTM cells were prepared using commercially available certified immortalized HTM cells (Applied Biological Materials Inc., Richmond, Canada) confirmed to be true HTM cells according to the criteria [20] described in a previous report [13,16]. Briefly, conventional 2D culture of the HTM cells was performed in the 2D culture medium composed of HG-DMEM containing 10% FBS, 1% l-glutamine, 1% antibiotic–antimycotic in 150 mm 2D culture dishes at 37 °C until reaching 90% confluence by changing the medium every other day. Thereafter, these HTM cells were further processed for 3D spheroid preparation by using a hanging droplet spheroid three-dimensional (3D) culture plate (#HDP1385, Sigma-Aldrich Co., St. Louis, MO, USA) for 6 days. Approximately 20,000 HTM cells were seeded in 28 μL of the 3D spheroid medium composed of the 2D culture medium supplemented by 0.25% methylcellulose in each well of the plate, and half of the medium was changed every following day. To replicate glaucomatous 3D HTM models, 250 nM DEX or 5 ng/mL TGF-β2 was supplemented on day 1 as described recently [16]. The rationale for the used concentrations of DEX and TGF-β2 were according to the response curve studies using different concentrations of DEX [21] and TGF-β2 [22]. 

### 2.2. Measurement of Real-Time Cellular Metabolic Functions

The oxygen consumption rate (OCR) and extracellular acidification rate (ECAR) of 5 ng/mL TGF-β2- or 250 nM DEX-treated or untreated 2D cultured HTM cells (as above) and those treated with 10 μM pan-ROCK-i ripasudil, and ROCK2-i KD025 for 24 h were each measured using a Seahorse XFe96 Bioanalyzer (Agilent Technologies, Santa Clara, CA, USA) according to the manufacturer’s instructions. In brief, 20 × 10^3^ 2D cultured cells per well were placed in a XFe96 Cell Culture Microplate (Agilent Technologies, #103794-100). After centrifugation of the plate at 1600 g for 10 min, the culture medium was replaced with 180 μL of the assay buffer (Seahorse XF DMEM assay medium (pH 7.4, Agilent Technologies, #103575-100) supplemented with 5.5 mM glucose, 2.0 mM glutamine, and 1.0 mM sodium pyruvate). The assay plates were incubated in CO_2_-free incubator at 37 °C for 1 h prior to the measurement. OCR and ECAR were measured in the Seahorse XFe96 Bioanalyzer under the 3 min mix and 3 min measure protocols at baseline and following injections of oligomycin (final concentration: 2.0 μM), carbonyl cyanide p-trifluoromethoxyphenylhydrazone (FCCP, final concentration: 5.0 μM), a mixture of rotenone/antimycin A (final concentration: 1.0 μM), and 2-deoxyglucose (2-DG, final concentration: 10 mM). 

### 2.3. Other Analytical Methods

3D HTM spheroids were prepared in the absence or presence of 250 nM DEX or 5 ng/mL TGF-β2 (as above), and those were compressed using a micro-squeezer (MicroSquisher, CellScale, Waterloo, ON, Canada) equipped with a microscale compression system composed of a 406 μm diameter cantilever for 10, 30, or 60 min as recently reported [13]. 

### 2.4. Quantitative PCR

Total RNA extraction, reverse transcription, real-time PCR, and quantification of the respective genes were performed as described previously [16]. As shown in Appendix A, qPCR of most of genes used Taqman probes except ER stress-related factors including glucose regulator protein (GRP) 78, GRP94, X-box binding protein 1 (XBP1), and spliced XBP1 (sXBP1) which were analyzed using the SYBR^®^ Green dye as reported previously [23].

All statistical analyses were performed using Graph Pad Prism 8 (GraphPad Software, San Diego, CA, USA) [16].

## 3. Results

### 3.1. Characterization of Real-Time Cellular Metabolic Functions of the TGF-β2- or DEX-Treated 2D and 3D Culture of HTM Cells in the Presence or Absence of ROCK Inhibitors (ROCK-Is)

In our recent study, we demonstrated that DEX or TGF-β2 resulted in mildly and severely downsized and stiff 3D HTM spheroids, respectively, thus making them viable in vitro HTM models for steroid-induced (SG) and primary open-angle glaucoma (POAG). That is, (1) DEX and TGF-β2 both caused a significant increase or decrease in the transendothelial electrical resistance (TEER) values and FITC dextran permeability as the barrier functions of the 2D HTM monolayer; (2) DEX or TGF-β2 induced mild and significant downsizing and an increase in stiffness, respectively, in the physical properties of the 3D HTM spheroids; (3) TGF-β2 and DEX induced significant and different alterations in the ECM gene expression, their modulators, and ER stress-related factors; (4) those characteristic properties were also diversely affected by pan-ROCK-i ripasudil, and ROCK2-i KD025 [18,24]. In the present study, to further characterize these models, we evaluated the real-time cellular metabolic functions of the 2D cultured HTM cells using a Seahorse analyzer. As shown in Figure 1, TGF-β2 significantly increased the glycolytic reserve of the ECAR and decreased the maximum respiration of the OCR, suggesting an energy shift from mitochondrial oxidative phosphorylation (OXPHOS) into glycolysis, and DEX induced similar but lesser effects. While in contrast, a substantial reverse energy shift from glycolysis into OXPHOS, that is, an increase in the spare respiratory reserve of the OCR and a decrease in the glycolytic reserve of the ECAR were observed in the presence of KD025, but not of Rip. Almost identical results were observed in the real-time cellular metabolic functions using the 3D HTM spheroids (Appendix A). These results indicated that (1) OXPHOS in glaucomatous HTM cells may be decreased, and (2) pharmacologically, ROCK2 or ROCK1 inhibition increases or decreases OXPHOS of HTM cells, suggesting that mitochondrial functions of the glaucomatous TM may be greatly affected by ROCK-is. 

### 3.2. Effects of Physical Compression Stresses on the 3D HTM Spheroids upon Continuous Mechanical Compression for 60 Min

Employing a 3D drop culture method [25], we established possible in vitro models of the SG or POAG TM (as above). However, as an important missing factor to be considered, we must take an additional factor to mimic the elevated IOP state into account. Therefore, to implement this study purpose, a mechanical compression stress was aimed at the 3D HTM spheroids. As shown in Figure 2, the 3D HTM spheroids were mechanically compressed to keep their deformation in the vertical semidiameter using a micro-squeezer system to evaluate physical stiffness of a single living 3D spheroid [13] for 10, 30, or 60 min. To study these mechanical compression stresses, mRNA expressions of the ECM proteins (collagen (COL) 1, 4, and 6, fibronectin (FN), and α-smooth muscle actin (SMA)), inflammatory cytokines (IL1B and IL6), and ER stress-related factors (glucose regulator protein (GRP) 78, GRP94, X-box binding protein 1 (XBP1), and spliced XBP1 (sXBP1)) were tested. Quite interestingly, the gene expressions of these molecules bimodally fluctuated until 60 min compression stresses. That is, during the 60 min compression stresses, most ECM proteins were (1) significantly upregulated during the initial 10 min, (2) substantially downregulated during the next 20 min, and (3) again significantly upregulated during the last 30 min (Figure 3). Similarly, the inflammatory cytokines and ER stress-related factors were also upregulated and downregulated during the initial 10 and the subsequent 20 min, respectively, and significantly upregulated during the last 30 min (Figure 3 and Figure 4). These collective data suggested that (1) the initial upregulation of most gene expressions evaluated (as above) may have been in response to the compression stresses, (2) their following downregulation may have been caused by adaptation to the compression stresses, and (3) their final substantial upregulation may have been caused by cellular stress and inflammation. In fact, significant upregulation of the gene expression of the inflammatory cytokines and most ER stress-related factors became evident during the last 30 min. 

### 3.3. Effects of Physical Compression Stresses on the 3D HTM Spheroids Treated with DEX or TGF-β2

Next, to further study the effects of physical compression stresses on the glaucomatous human TM, we evaluated the effects of the mechanical compression stresses for 30 min, in which the 3D HTM spheroids were relatively stable against these compression stresses as compared to other phases, in which they greatly fluctuated, on the mRNA expression of the ECM proteins, inflammatory cytokines, and ER stress-related factors (as above) in the 3D HTM spheroids treated with 5 ng/mL TGF-β2 or 250 ng/mL DEX. As shown in Figure 5 and Figure 6, such mechanical compressions induced a substantial downregulation of most genes tested, that is, (1) a significant upregulation of COL1 and downregulation of αSMA, IL1, GRP78, and tXBP or (2) a significant downregulation of COL6, αSMA, IL1, IL6, GRP78, and tXBP in the TGF-β2- or DEX-treated 3D HTM spheroids, respectively, as compared to the untreated 3D spheroids. Therefore, these data suggested that glaucomatous HTM models, TGF-β2- or DEX-treated 3D HTM spheroids may have already been resistant to mechanical compression stresses toward inflammatory cytokines and ER stress-related factors.

## 4. Discussion

In terms of the primary etiology of the GON, axonal damage at the optic nerve head (ONH) causing chronic retinal ganglion cell (RGC) apoptosis has been suggested [26,27]. Similarly, Leber’s hereditary optic neuropathy (LHON) is well-known to manifest subacute RGC apoptosis as well as axonal injury at the OHN caused by mitochondrial abnormalities [28,29]. Based upon such analogy between the GON and LHOH, to investigate a possible relationship between mitochondrial abnormalities and the GON, Abu-Amero et al. sequenced the entire mitochondrial (mt)DNA coding region in 27 patients with POAG [30]. The results revealed that 27 unique nonsynonymous mtDNA changes were detected in the patients with POAG despite only three benign polymorphisms identified in MYOC and OPTN in both patients with POAG and the control subjects. In addition, a significant decrease in the mitochondrial respiratory activity was observed in the patients with POAG as compared with the control subjects (*p* < 0.001). Therefore, they rationally concluded that mitochondrial abnormalities implicating oxidative stress may cause mitochondria dysfunction in patients with POAG [30]. Such possible mechanisms related by mitochondrial dysfunction were also suggested in patients with primary congenital glaucoma [31]. In fact, several studies demonstrated that pathological molecular events within the GON are related to the long-term accumulation of oxidative damage arising from mitochondrial dysfunctions [32,33]. In the current study, we also found a decrease in OXPHOS in our in vitro models of POAG and SG using TGF-β2 and DEX-treated HTM cells using a real-time cellular metabolic functions analyzer, providing additional rationale for these in vitro glaucomatous TM models. Furthermore, ROCK2 inhibition by KD025, but not pan-ROCK inhibition by Rip induced a significant stimulation of OXPHOS in the HTM cells in the absence or presence of TGF-β2 or DEX. Therefore, this methodology can be beneficial for pharmacological evaluations of the mitochondrial functions of HTM cells as well as for estimating drug-induced effects of other antiglaucoma medications toward HTM cells.

In our recent study, to establish suitable in vitro models replicating HTM with POAG and SG, we employed a unique 3D spheroid drop culture method [13,14,15] for the engineering of intraocular tissues which serve as suitable in vitro disease models for a better understanding of various diseases [25,34] in addition to the conventional 2D culture [16,17,18]. As for the 3D cell culture of HTM cells, several alternative methods using scaffold-assisted [35] and polymer gels [36] have been developed to replicate the glaucomatous HTM architecture. As compared with these methods, our method did not require unnecessary staffs, such as scaffold or gel polymers that are not present within the physiological conditions of the human TM, and, therefore, we could isolate a single living 3D spheroid to directly measure the physical properties, size, and stiffness (as above) [13,14,15]. In the current study, using this advantage, we applied direct mechanical compression stresses toward living 3D HTM spheroids, which may reproduce the mechanical compression toward HTM cells caused by elevated IOP. Upon such direct compression stresses, we observed bimodal fluctuation of the mRNA expression of the ECM proteins for 60 min; (1) significant upregulations during the initial 10 min, (2) substantial downregulation during the subsequent 20 min, and (3) again, significant upregulation for the last 30 min. Similarly, the mRNA expression of the inflammatory cytokines and ER stress-related factors were also bimodally changed. Furthermore, such subacute compression stresses toward the TGF-β2- or DEX-treated 3D HTM spheroids induced much fewer effects toward the mRNA expression of several tested molecules, that is, upregulation of COL1 and downregulation of FN, as well as downregulation of most inflammatory cytokines and ER stress-related factors as compared with those of the untreated control 3D spheroids. These collective results indicated that (1) our current in vitro 3D HTM spheroids compression models may reflect acute, subacute, and chronic responses of HTM cells toward mechanical compression stresses such as elevated IOP, and (2) TGF-β2- or DEX-treated 3D HTM spheroids may already be resistant toward mechanical compression stresses.

As study limitations to this study, currently, the precise mechanisms causing the bimodal fluctuation of the mRNA expression of ECM proteins and changes in ECM protein amounts during continuous compression stresses as above remain to be elucidated, although those were quite reproducible. Furthermore, as another important issue, the response of HTM cells to steroids may differ, since clinically, it is well-known the presence of steroid responders accounts for a small percentage of the overall population. Furthermore, there are at least more than 100 genes are identified to be associated with glaucoma etiology [37,38]. Thus, additional studies to elucidate the currently unidentified biological mechanisms responsible for causing such bimodality upon compression stress and differences in the response of HTM cells to steroids using RNA sequence analysis, metabolome analysis, and others using additional in vivo experimental conditions will be needed to elucidate unknown mechanisms as above and such studies are our next projects.

## Figures and Tables

**Figure 1 biomedicines-10-01338-f001:**
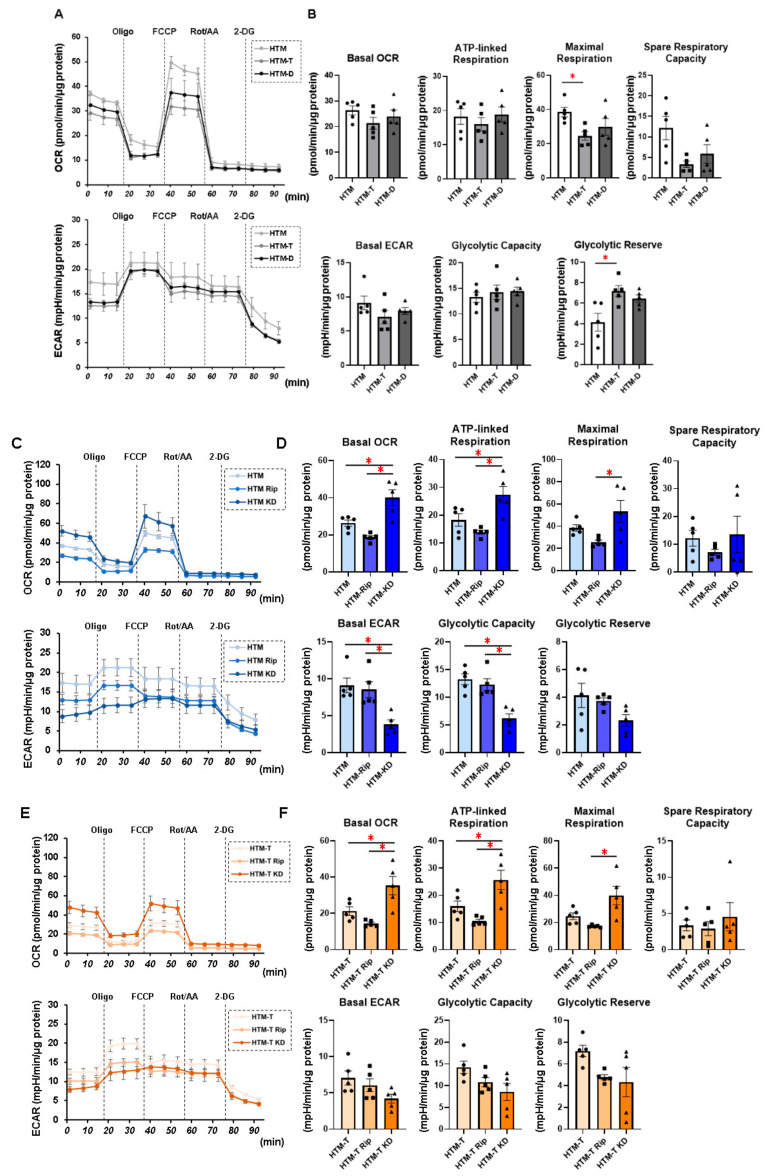
Measurement of mitochondrial and glycolytic functions of TGF-β2- or DEX-treated or untreated 2D cultured HTM cells in the absence or presence of ROCK inhibitors (ROCK-is).Under the following four conditions: (1) 2D cultured HTM cells (HTM) and those treated with 5 ng/mL TGF-β2 (HTM-T) or 250 nM DEX (HTM-D), (2) untreated 2D cultured HTM cells treated without (HTM) or with 10 μM pan-ROCK-i ripasudil (HTM Rip) or ROCK2-i KD025 (HTM KD) for 24 h, (3) 5 ng/mL TGF-β2 (TGF)-treated 2D cultured HTM cells treated without (HTM-T) or with 10 μM pan-ROCK-i ripasudil (HTM-T Rip) or ROCK2-i KD025 (HTM-T KD) for 24 h, and (4) 250 nM DEX-treated 2D cultured HTM cells treated without (HTM-D) or with 10 μM pan-ROCK-i ripasudil (HTM-D Rip) or ROCK2-i KD025 (HTM-D KD) for 24 h, 2D cultured HTM cells were subjected to mitochondrial and glycolysis function analyses using a Seahorse XFe96 Bioanalyzer. Measurements of the oxygen consumption rate (OCR) and extracellular acidification rate (ECAR) before drug injections (at baseline) were calculated as 100%, and their changes were determined by the following injections: (i) oligomycin (a complex V inhibitor), (ii) FCCP (a protonophore), (iii) rotenone/antimycin (complex I/III inhibitors), and (iv) 2-DG (a hexokinase inhibitor). (**A**,**C**,**E**,**G**) Measurements of the OCR and ECAR. (**B**,**D**,**F**,**H**) Plots for subcomponents of OCR (basal, ATP-linked respiration, maximum respiration, and spare respiratory capacity) and ECAR (basal, glycolytic capacity, and glycolytic reserve). ATP-linked respiration was defined as the average OCR in the presence of oligomycin. Maximum respiration was defined as the average OCR in the presence of FCCP. Spare respiratory reserve was defined as the difference between the average OCR in the presence of FCCP and the average OCR at baseline. Glycolytic reserve was defined as the difference between the final measurement of the ECAR in the presence of oligomycin and the average ECAR. The data are presented as the means ± the standard error of the mean (SEM). Note: * *p* < 0.05 (Student’s *t*-test).

**Figure 2 biomedicines-10-01338-f002:**
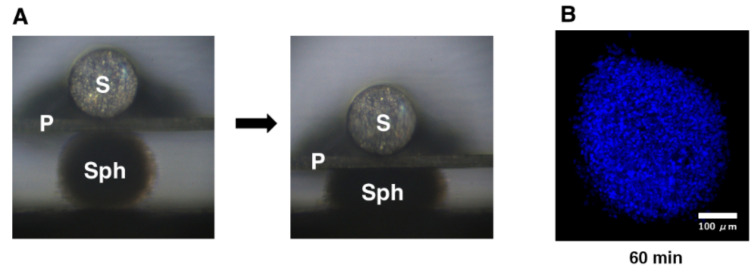
Mechanical compression toward 3D HTM spheroids. (**A**) Thirty 3D HTM spheroids were mechanically compressed using a micro-squeezer system (Sph: 3D sphenoid, S: pressure sensor, P: compression plate) for 10, 30, or 60 min. (**B**) Representative immunolabeling image of a 3D HTM spheroid mechanically compressed for 60 min stained with DAPI (scale bar: 100 µm).

**Figure 3 biomedicines-10-01338-f003:**
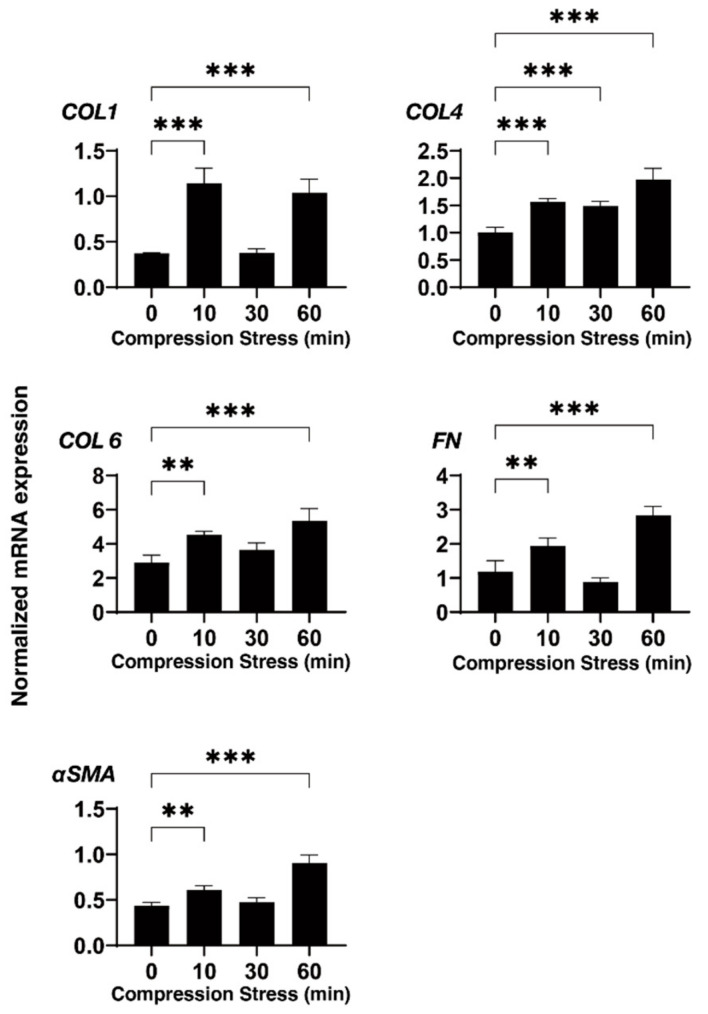
mRNA expression of the ECM proteins of the 3D HTM spheroids after a mechanical compression stress for different time durations. Untreated 3D HTM spheroids were mechanically compressed on day 6 for 10, 30, or 60 min and subjected to a qPCR analysis to estimate the expression of mRNA of the ECM proteins (*COL 1*, *COL 4*, *COL 6*, *FN*, and *a-SMA*). All the experiments were performed in duplicate using fresh preparations (*n* = 4). The data are presented as the arithmetic means ± standard error of the mean (SEM). Note: ** *p* < 0.01, *** *p* < 0.005 (ANOVA followed by Tukey’s multiple comparison test).

**Figure 4 biomedicines-10-01338-f004:**
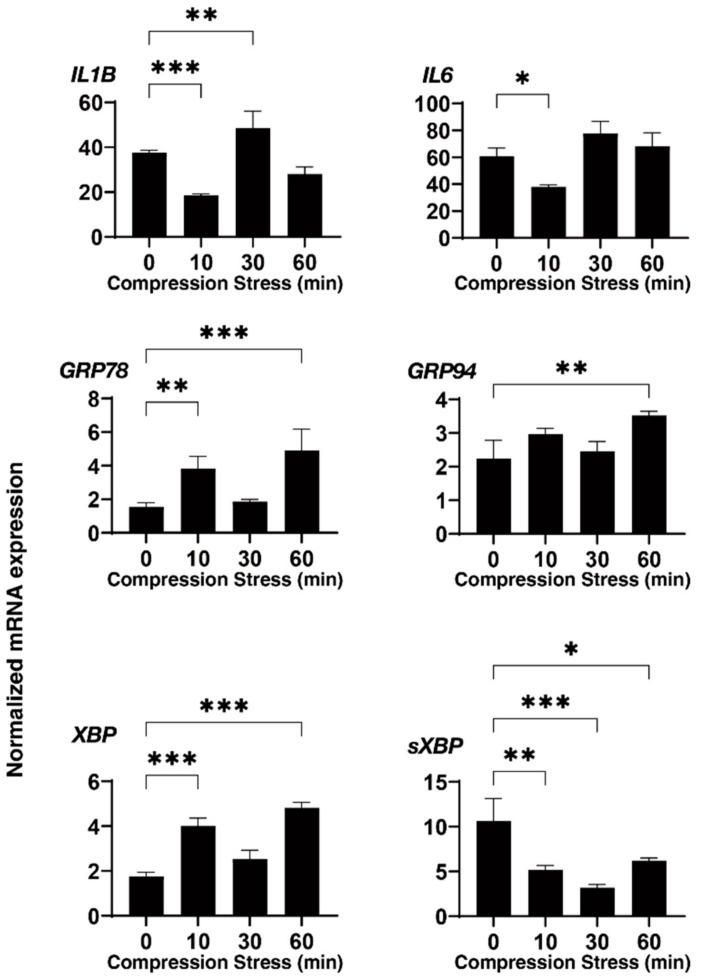
mRNA expression of the inflammatory cytokines and ER stress-related factors of the 3D HTM spheroids after a mechanical compression stress for different time durations. Untreated 3D HTM spheroids were mechanically compressed on day 6 for 10, 30, or 60 min and subjected to a qPCR analysis to estimate the expression of mRNA of the inflammatory cytokines (IL1 and IL6) and ER stress-related factors (glucose regulator protein (GRP) 78, GRP94, X-box binding protein 1 (XBP1), and spliced XBP1 (sXBP1)). All the experiments were performed in duplicate using fresh preparations (*n* = 4). The data are presented as the arithmetic means ± standard error of the mean (SEM). Note: * *p* < 0.05, ** *p* < 0.01, *** *p* < 0.005 (ANOVA followed by Tukey’s multiple comparison test).

**Figure 5 biomedicines-10-01338-f005:**
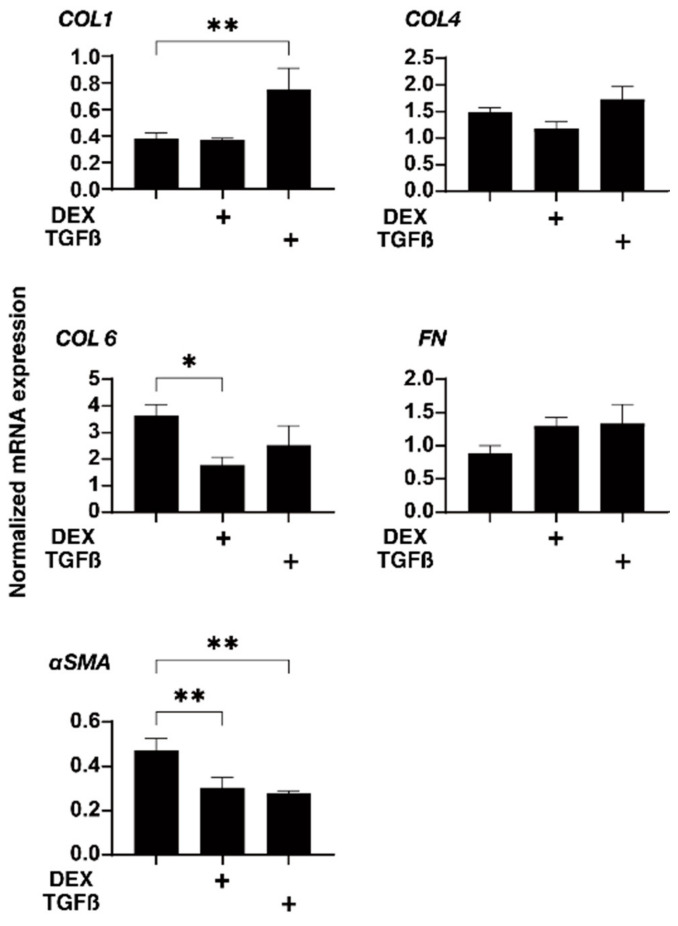
mRNA expression of the ECM proteins of the 3D HTM spheroids after a mechanical compression stress for 30 min. In the presence of 5 ng/mL TGF-β2 (TGF) or 250 ng/mL DEX, the 3D HTM spheroids were mechanically compressed on day 6 for 30 min and subjected to a qPCR analysis to estimate the expression of mRNA of the ECM proteins (*COL 1*, *COL 4*, *COL 6*, *FN*). These data were compared with those of the untreated 3D HTM spheroids compressed for 30 min (as above). All the experiments were performed in duplicate using fresh preparations (*n* = 4). The data are presented as the arithmetic means ± standard error of the mean (SEM). Note: * *p* < 0.05, ** *p* < 0.01, (ANOVA followed by Tukey’s multiple comparison test).

**Figure 6 biomedicines-10-01338-f006:**
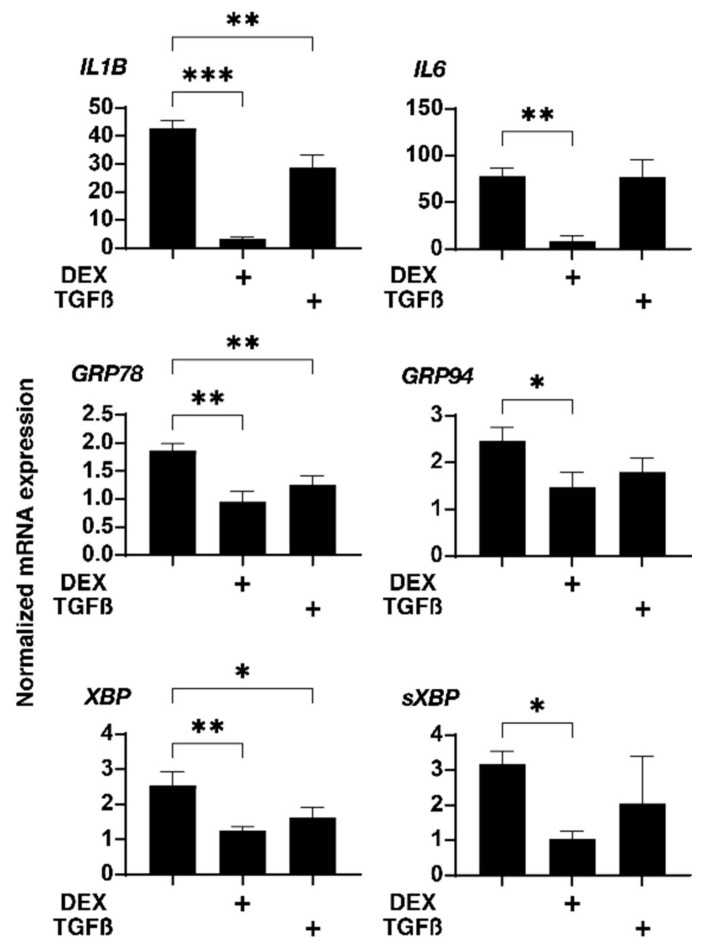
mRNA expression of the inflammatory cytokines and ER stress-related factors of the 3D HTM spheroids after a mechanical compression stress for 30 min. In the presence of 5 ng/mL TGF-β2 (TGF) or 250 ng/mL DEX, the 3D HTM spheroids were mechanically compressed on day 6 for 30 min and subjected to a qPCR analysis to estimate the expression of mRNA of the inflammatory cytokines (IL1 and IL6) and ER stress-related factors (glucose regulator protein (GRP) 78, GRP94, X-box binding protein 1 (XBP1), and spliced XBP1 (sXBP1)). These data were compared with those of the untreated 3D HTM spheroids compressed for 30 min (as above). All the experiments were performed in duplicate using fresh preparations (*n* = 4). The data are presented as the arithmetic means ± standard error of the mean (SEM). Note: * *p* < 0.05, ** *p* < 0.01, *** *p* < 0.005 (ANOVA followed by Tukey’s multiple comparison test).

## Data Availability

The data that support the findings of this study are available from the corresponding author upon reasonable request.

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
