# Peer review of "Human Trabecular Meshwork (HTM) Cells Treated with TGF-β2 or Dexamethasone Respond to Compression Stress in Different Manners"

_biomedicines, 2022, doi:10.3390/biomedicines10061338_

Round 1

Reviewer 1 Report

This manuscript by Watanabe and colleagues investigate the different effect of TGF-beta 2 and Dex on 2D and 3D models of HTM cells upon mechanical compression stresses. I have some comments that need to be addressed.

  1. It is well established that that genes exert their functions via their relevant protein expression including ECM proteins, and it is also well known that the mRNA expression level of many genes are not correlated with their protein expression. Generally, The turnover rate or half-life of ECM proteins are very long. However, the authors used a very short period of time windows (from 30-60 mins or 60 mins) to monitor the ECM gene expression changes. There is some serious flaw on this, because it is reasonable to expect that the ECM proteins should not be significantly changed in such a short period of time. I would like the authors to check the ECM protein changes after treatment in this study.

  1. Another issue is I am not sure whether TGF-beta2 or DEX can penetrate to the every cell of the 3D culture of HTM cells equally, because the 3D spheroid is relatively big and there may be some related barriers there. Please provide evidence to support it.

  1. The authors used the real time QPCR, I would like the authors to show the standard cures of each genes and the linearity analyses data. In addition, please add the length of PCR products.

  1. From the supplementary data, it showed that the primers of GRP78,GRP94,XBP and SXBP don’t contain probes compared with other gens, indicating that different QPCR procedure may be used. Please give a detailed information regarding the QPCR procedure.

  1. I was also wondering whether the TGF-beta 2 or DEX treatment also change the total ATP concentration.?

  1. I was also wondering whether “gene expression of ECM proteins” is OK, generally, we use “ECM gene expression”.

  1. From 3.2., please change “continu ous” to “continuous”

Thanks for the invitation.

Author Response

Dear Editor,

Thank you very much for the constructive comments concerning our manuscript, " Human trabecular meshwork (HTM) cells treated with TGF-b2 or dexamethasone respond to compression stress in different manners”. We examined the Reviewer's comments carefully and prepared a revised version of our paper that takes these comments into account. The changes are listed below.

Editorial comments

As pointed out, almost identical phrase marked by highlights were rewritten.

Reviewer 1

This manuscript by Watanabe and colleagues investigate the different effect of TGF-beta 2 and Dex on 2D and 3D models of HTM cells upon mechanical compression stresses. I have some comments that need to be addressed.

  1. It is well established that that genes exert their functions via their relevant protein expression including ECM proteins, and it is also well known that the mRNA expression level of many genes are not correlated with their protein expression. Generally, The turnover rate or half-life of ECM proteins are very long. However, the authors used a very short period of time windows (from 30-60 mins or 60 mins) to monitor the ECM gene expression changes. There is some serious flaw on this, because it is reasonable to expect that the ECM proteins should not be significantly changed in such a short period of time. I would like the authors to check the ECM protein changes after treatment in this study.

Answer; Thank you for this comment. I totally agree that ECM protein turnover should be more longer that 60 mins, and therefore, I do not assume that ECM protein changes may be not significantly changed during 60 mins. However, I think that the important issue is to the changes of gene expressions of ECM and other genes rather than those protein changes, because timing of the gene expression and protein expression should be different in the case of acute elevation of the intraocular pressure. In addition, I am afraid that more longer exposure to the continued compression stresses may lead cytotoxic effects, and therefore, we performed until 60 mins. However, as suggested, I agree that the protein expression changes of ECM still important points, and therefore, this information is included as the study limitation in the last paragraph of discussion; ” As study limitations to this study, currently the precise mechanisms causing the bimodal fluctuation of the mRNA expressions of ECM proteins and changes of ECM proteins amounts during the continued compression stresses as above were remain to be elucidated, although those were quite reproducible. Furthermore, as another important issue, the response of HTM cells to steroid may different, since clinically, it is well known the presence of steroid responders account for a small percentage of populations. Thus, additional studies to elucidate unidentified biological mechanisms causing such bimodality upon compression stresses and difference in the response of HTM cells to steroid using RNA sequence analysis, metabolome analysis, and others using additional in vivo experimental conditions as our next projects.”.

  1. Another issue is I am not sure whether TGF-beta2 or DEX can penetrate to the every cell of the 3D culture of HTM cells equally, because the 3D spheroid is relatively big and there may be some related barriers there. Please provide evidence to support it.

Answer; Thank for this comment. In terms of penetration of TGF-beta2 or DEX into each cells of the 3D spheroids, I do not know exactly. Nevertheless, under current 3D spheroid cell culture system, no apoptosis cells were not present within the inside of the 3D spheroid based upon DAPI staining in our previous study. Therefore, this strongly indicated that at least, oxygen and culture medium including testing drugs should be penetrated into the inside of the 3D spheroid.

  1. The authors used the real time QPCR, I would like the authors to show the standard cures of each genes and the linearity analyses data. In addition, please add the length of PCR products.

Answer; As suggested, the standard curves of each genes, the linearity analyses data (Supplemental Fig. 2) and the length of PCR products (Supplemental Table 1) are included as the supplemental materials.

  1. From the supplementary data, it showed that the primers of GRP78,GRP94,XBP and SXBP don’t contain probes compared with other gens, indicating that different QPCR procedure may be used. Please give a detailed information regarding the QPCR procedure.

Answer; Thank you for this comment. As pointed out, qPCR analysis of the ER stress related genes was performed using SYBR Green and others were performed using Tagman qPCR system because both were established quite reproducible and reliable in our previous studies. Therefore, the detailed information related to the qPCR methods were included; “Total RNA extraction, reverse transcription, rea

l-time PCR, and quantification of the respective genes were as described previously[16]. As shown in Supplemental Tables 1 and 2, qPCR of the most of genes were using Taqman probes except ER stress related factors including the glucose regulator protein (GRP)78, GRP94, the X-box binding protein-1 (XBP1), and spliced XBP1 (sXBP1) which were analyzed using SYBR® Green dye as according previously [23].”, and Supplemental Table 2. 

  1. I was also wondering whether the TGF-beta 2 or DEX treatment also change the total ATP concentration.?

Answer; Thank you so much for this interesting comment. Unfortunately, I understand that this issue is important, but we can not measure the total ATP concentrations in case of the TGF-b or DEX treatment conditions by Seahorse analysis. Nevertheless, I assume that those total ATP concentrations among several conditions may not be so different with each other because it is known that the total ATP concentrations are critically regulated (Nat Rev Mol Cell Biol 2012, 13 (4), 251.).

  1. I was also wondering whether “gene expression of ECM proteins” is OK, generally, we use “ECM gene expression”.

Answer; Thank you for this comment. As pointed out, “gene expression of ECM proteins” was changed to “ECM gene expression” in the Abstract.

  1. From 3.2., please change “continu ous” to “continuous”

Answer; Thank you for this comment. As pointed out, “continu ous” was corrected to “continuous”

Reviewer 2

The study demonstrates the cellular responses of the TGF-β2 or dexamethasone treated or untreated 3D human trabecular meshwork (HTM) cells model upon mechanical compression stresses by real time cellular metabolism analyzer. There are some issues should be clarified.

  1. In real world, steroid responders account for a small percentage of populations. Therefore, the response of trabecular meshwork (HTM) cells to steroid may different.

Answer; Thank you for this comment. I agree that if difference in the response of HTM may be different, those will become an important clue for better understanding the molecular mechanisms of steroid responder. Therefore this information is included in the discussion; “As study limitations to this study, currently the precise mechanisms causing the bimodal fluctuation of the mRNA expressions of ECM proteins and changes of ECM proteins amounts during the continued compression stresses as above were remain to be elucidated, although those were quite reproducible. Furthermore, as another important issue, the response of HTM cells to steroid may different, since clinically, it is well known the presence of steroid responders account for a small percentage of populations. Thus, additional studies to elucidate unidentified biological mechanisms causing such bimodality upon compression stresses and difference in the response of HTM cells to steroid using RNA sequence analysis, metabolome analysis, and others using additional in vivo experimental conditions as our next projects.”  

  1. Why choose the dose 250 nM DEX or or 5 ng/mL TGFβ2? Is there any dose response curve study?

Answer; Thank you for this comment. As suggested, to prove rationale for used concentrations of DEX and TGFb2, response curve studies using different concentrations were included; DEX (Wang C, Li L, Liu Z. Experimental research on the relationship between the stiffness and the expressions of fibronectin proteins and adaptor proteins of rat trabecular meshwork cells. BMC Ophthalmol. 2017 Dec 29;17(1):268. doi: 10.1186/s12886-017-0662-5.) and TGFb2 (Igarashi N, Honjo M, Yamagishi R, Kurano M, Yatomi Y, Igarashi K, Kaburaki T, Aihara M. Crosstalk between transforming growth factor β-2 and Autotaxin in trabecular meshwork and different subtypes of glaucoma. J Biomed Sci. 2021 Jun 17;28(1):47. doi: 10.1186/s12929-021-00745-3.). These information are included in the Method; “2D and 3D spheroid cultures of the HTM cells were prepared as described in a previous report [13,16]. Briefly, conventional 2D cultures of the HTM cells were performed in a 2D culture medium composed of HG-DMEM containing 10 % FBS, 1 % L-glutamine, 1 % antibiotic-antimycotic in 150 mm 2D culture dishes at 37°C until reaching 90 % confluence by changing the medium every other day. Thereafter, these HTM cells were further processed to produce a 3D spheroid preparation by using a hanging droplet spheroid three-dimension (3D) culture plate (# HDP1385, Sigma-Aldrich) for a period of 6 days. Approximately 20,000 HTM cells were seeded in 28 ml of the 3D spheroid medium composed of the 2D culture medium supplemented with 0.25 % methylcellulose in each well of the plate, and half of the medium was each exchanged daily. To replicate glaucomatous 3D HTM models, 250 nM DEX or 5 ng/mL TGFβ2 was supplemented at Day 1 as described recently [16]. The rationale for the used concentrations of DEX and TGFb2 were according response curve studies using different concentrations of DEX [21] and TGFb2 [22].”.

  1. Why the bimodal fluctuation of the mRNA expressions of ECM proteins, i.e. initial significantly up-regulations (0-10 mins), subsequent down-regulations (10- 30 mins) and following up-regulation again (30-60 mins) can represent acute, subacute and chronic HTM models affected by elevated intraocular pressures?

Answer; Thank you for this comment. In terms of the bimodal fluctuation of the mRNA expressions of ECM proteins during the course of continuous compression stresses, to be honest, we were very surprised and unexpected experimental observation, although this was quite reproducible. Therefore, as present, the precise mechanisms causing such bimodality were still be unidentified. Therefore, this information is included as the study limitation in the last paragraph of discussion; “As study limitations to this study, currently the precise mechanisms causing the bimodal fluctuation of the mRNA expressions of ECM proteins and changes of ECM proteins amounts during the continued compression stresses as above were remain to be elucidated, although those were quite reproducible. Furthermore, as another important issue, the response of HTM cells to steroid may different, since clinically, it is well known the presence of steroid responders account for a small percentage of populations. Thus, additional studies to elucidate unidentified biological mechanisms causing such bimodality upon compression stresses and difference in the response of HTM cells to steroid using RNA sequence analysis, metabolome analysis, and others using additional in vivo experimental conditions as our next projects.”

Reviewer 2 Report

The study demonstrates the cellular responses of the TGF-β2 or dexamethasone treated or untreated 3D human trabecular meshwork (HTM) cells model upon mechanical compression stresses by real time cellular metabolism analyzer. There are some issues should be clarified.

  1. In real world, steroid responders account for a small percentage of populations. Therefore, the response of trabecular meshwork (HTM) cells to steroid may different.
  2. Why choose the dose 250 nM DEX or or 5 ng/mL TGFβ2? Is there any dose response curve study?
  3. Why the bimodal fluctuation of the mRNA expressions of ECM proteins, i.e. initial significantly up-regulations (0-10 mins), subsequent down-regulations (10- 30 mins) and following up-regulation again (30-60 mins) can represent acute, subacute and chronic HTM models affected by elevated intraocular pressures?

Author Response

(The authors gave the same response as above.)

Round 2

Reviewer 1 Report

The authors addressed some of the concerns, but raise more questions regarding the quality of the work. 

1. From the supplementary table 1 regarding the length of the PCR products, the authors listed the PCR products lengths as the following numbers: 4413, 17554,158195, 23279, 75204,56324, 6491, 17517, 6010, 6010, 7029, 4799.  It did not make sense, the PCR product lengths would be ranged from less than 100 bps to less than 200 bps in the real time qPCR run. For example, the PCR product length of the first gene listed in the table 1 (named as human RPLP0) should be 143bp after Blast search analyses of the sense and antisense primers. I really didn’t understand how can the authors get the PCR product length of human RPLP0 at 4413 bp.  Some PCR product length even reaches more than 70000 bp. Please recheck all the PCR product length carefully.

2.  The authors acknowledge that the ECM protein may not change under such short period of time treatment condition.  There are more easy techniques to use than the qPCR for analyzing  gene expression or transcriptional changes using RNA sequencing. The authors at least to discuss it in their manuscript.

3. There are more than 100 genes are associated with glaucoma, so the authors should   discuss the possibilities that the TGF beta or DEX  treatments may also affect these glaucoma-related gene changes.

Author Response

Dear Editor,

Thank you very much for the constructive comments concerning our manuscript, " Human trabecular meshwork (HTM) cells treated with TGF-b2 or dexamethasone respond to compression stress in different manners”. We examined the Reviewer's comments carefully and prepared a revised version of our paper that takes these comments into account. The changes are listed below.

Reviewer 1

The authors addressed some of the concerns, but raise more questions regarding the quality of the work. 

  1. From the supplementary table 1 regarding the length of the PCR products, the authors listed the PCR products lengths as the following numbers: 4413, 17554,158195, 23279, 75204,56324, 6491, 17517, 6010, 6010, 7029, 4799.  It did not make sense, the PCR product lengths would be ranged from less than 100 bps to less than 200 bps in the real time qPCR run. For example, the PCR product length of the first gene listed in the table 1 (named as human RPLP0) should be 143bp after Blast search analyses of the sense and antisense primers. I really didn’t understand how can the authors get the PCR product length of human RPLP0 at 4413 bp.  Some PCR product length even reaches more than 70000 bp. Please recheck all the PCR product length carefully.

Answer; Thank you for this comment. I apologize my previous answer because I was misunderstood and miscalculated the PCR products. Therefore, I again calculated using Blast search analyses as suggested, and therefore corresponding supplemental Table 1 was corrected.

  1. The authors acknowledge that the ECM protein may not change under such short period of time treatment condition.  There are more easy techniques to use than the qPCR for analyzing gene expression or transcriptional changes using RNA sequencing. The authors at least to discuss it in their manuscript.

Answer; Thank you so much for this comment. I agree that RNA sequence analysis will be very powerful strategy to identify unknown additional mechanisms. Therefore, this information is included within the study limitation in the Discussion;” As study limitations to this study, currently the precise mechanisms causing the bimodal fluctuations of the mRNA expressions of ECM proteins and changes in the levels of ECM proteins during the continued compression stress as above remain to be elucidated, although the results were quite reproducible. Another important issue is, the response of HTM cells to steroids may be different, since clinically, it is well known the presence of steroid responders account for only a small percentage of the overall population. Furthermore, there are at least more than 100 genes are identified to be associated with glaucoma etiology [37][38]. Thus, additional studies to elucidate the currently unidentified biological mechanisms responsible for causing such bimodality upon compression stress and differences in the response of HTM cells to steroids using RNA sequence analysis, metabolome analysis, and others using additional in vivo experimental conditions will be needed to elucidate unknown mechanisms as above and such studies are our next projects.”

  1. There are more than 100 genes are associated with glaucoma, so the authors should  discuss the possibilities that the TGF beta or DEX  treatments may also affect these glaucoma-related gene changes.

Answer; Thank you so much for this comment. I agree that it will be very interesting to identify the master regulator genes among the glaucoma related genes as suggested, therefore as suggested as above #2 question, I would like to perform RNA sequence analysis to identify such the master genes. Therefore, this information is included within the study limitation in the Discussion; “As study limitations to this study, currently the precise mechanisms causing the bimodal fluctuations of the mRNA expressions of ECM proteins and changes in the levels of ECM proteins during the continued compression stress as above remain to be elucidated, although the results were quite reproducible. Another important issue is, the response of HTM cells to steroids may be different, since clinically, it is well known the presence of steroid responders account for only a small percentage of the overall population. Furthermore, there are at least more than 100 genes are identified to be associated with glaucoma etiology [37][38]. Thus, additional studies to elucidate the currently unidentified biological mechanisms responsible for causing such bimodality upon compression stress and differences in the response of HTM cells to steroids using RNA sequence analysis, metabolome analysis, and others using additional in vivo experimental conditions will be needed to elucidate unknown mechanisms as above and such studies are our next projects.”.

Reviewer 2

The authors have revised the manuscript according to my suggestions. I have no more comments.

Answer; Thank you so much for this comment.

Reviewer 2 Report

The authors have revised the manuscript according to my suggestions. I have no more comments.

Author Response

(The authors gave the same response as above.)

Round 3

Reviewer 1 Report

I have checked the PCR products length again, it appears that many revised numbers are still wrong, it raises an issue whether  the authors know how to run qPCR and calculate the predicted PCR products length.

Author Response

Dear Editor,

Thank you very much for the constructive comments concerning our manuscript, " Human trabecular meshwork (HTM) cells treated with TGF-b2 or dexamethasone respond to compression stress in different manners”. We examined the Reviewer's comments carefully and prepared a revised version of our paper that takes these comments into account. The changes are listed below.

Reviewer 1

The authors addressed some of the concerns, but raise more questions regarding the quality of the work. 

  1. I have checked the PCR products length again, it appears that many revised numbers are still wrong, it raises an issue whether  the authors know how to run qPCR and calculate the predicted PCR products length.

Answer; Thank you for this comment. I apologize my previous answer because I was misunderstood and miscalculated the PCR products. Therefore, I again calculated using NCBI (Nuculeotide) search analyses as suggested, and therefore corresponding supplemental Table 1 was corrected. The table of calculation is shown.

Primer1(Fw)
position

Primer2(Rv)
position

Primer2(Rv)
count

product length (bp)

36B4

157

35

21

143

COL1

244

133

20

131

COL4

5132

5038

23

117

COL6

373

252

20

141

FN

699

606

21

114

SMA

893

785

22

130

GRP78

481

397

20

104

GRP94

1290

1096

23

217

XBP

265

134

21

152

sXBP

662

370

21

313

IL1B

136

52

21

105

IL6

403

301

18

120
